# Preparation and Evaluation of pH-Sensitive Chitosan/Alginate Nanohybrid Mucoadhesive Hydrogel Beads: An Effective Approach to a Gastro-Retentive Drug Delivery System

**DOI:** 10.3390/pharmaceutics16111451

**Published:** 2024-11-13

**Authors:** Sadia Rehman, Qazi Adnan Jamil, Sobia Noreen, Muhammad Azeem Ashraf, Asadullah Madni, Hassan Mahmood, Hina Shoukat, Muhammad Rafi Raza

**Affiliations:** 1Department of Pharmaceutics, Faculty of Pharmacy, The Islamia University of Bahawalpur, Bahawalpur 63100, Pakistan; sadia.rehman@riphahsahiwal.edu.pk (S.R.); asadullah.madni@iub.edu.pk (A.M.); Hinashoukat50@yahoo.com (H.S.); 2Department of Pharmaceutical Technology, Institute of Pharmacy, University of Innsbruck, Innrain 52, 6020 Innsbruck, Austria; 3Department of Supply Chain, University of Management and Technology Lahore, Lahore 54770, Pakistan; azeem.ashraf@ymail.com; 4Linguistics & Literature Department, COMSATS University Islamabad, Lahore Campus, Lahore 54000, Pakistan; hamehmood05@gmail.com; 5Department of Mechanical Engineering, COMSATS University Islamabad, Sahiwal Campus, Sahiwal 57000, Pakistan; rafirazamalik@cuisahiwal.edu.pk

**Keywords:** graphene oxide (GO), hydrogel beads, chitosan (CS), folic acid (Fa), mucoadhesive

## Abstract

Background: Despite extensive research over the decades, cancer therapy is still a great challenge because of the non-specific delivery of chemotherapeutic agents, which could be overcome by limiting the distribution of chemotherapeutic agents toward cancer cells. Objective: To reduce the cytolytic effects against cancer cells, graphene oxide (GO) nanoparticles (NPs) can load anticancer medicines and genetic tools. Methodology: During the current study, folic-acid-conjugated graphene oxide (Fa-GO) hybrid mucoadhesive chitosan (CS)-based hydrogel beads were fabricated through an “ion-gelation process”, which allows for regulated medication release at malignant pH. Results: The fabricated chitosan–alginate (SA-CS) hydrogel beads were examined using surface morphology, optical microscopy, XRD, FTIR, and homogeneity analysis techniques. The size analysis indicated that the size of the Fa-GO was up to 554.2 ± 95.14 nm, whereas the beads were of a micrometer size. The folic acid conjugation was confirmed by NMR. The results showed that the craggy edges of the graphene oxide were successfully encapsulated in a polymeric matrix. The mucoadhesive properties were enhanced with the increase in the CS concentration. The nanohybrid SA-CS beads exhibited good swelling properties, and the drug release was 68.29% at pH 5.6 during a 24 h investigation. The accelerated stability study, according to ICH guidelines, indicated that the hydrogel beads have a shelf-life of more than two years. Conclusions: Based on the achieved results, it can be concluded that this novel gastro-retentive delivery system may be a viable and different way to improve the stomach retention of anticancer agents and enhance their therapeutic effectiveness.

## 1. Introduction

Cancer is characterized by the uncontrolled multiplication and metastasis of abnormal cells. The high incidence and fatality rate of cancer make it a major disease that seriously endangers human health [1,2]. Factors including *Helicobacter pylori* infection, tobacco use, alcoholic beverages, excessive body fat, and age-related mental decline have all been linked to the development of gastric cancer [3]. An alarming number of people throughout the world are being diagnosed with gastric cancer, an aggressive illness linked to severe complications [4,5]. Even though researchers have made tremendous progress in developing alternatives to surgery for stomach cancer, more work is needed, particularly in the area of gastric-targeted drug delivery [6]. Chemotherapy is now utilized to treat cancer because cytotoxic medications are considered promising anticancer agents. These treatments alter cancer cells’ ability to stop their cell cycle by creating free radicals, which increases the apoptosis rate [7]. Doxorubicin HCl (Dox.H), a four-ring molecule from the anthracycline antibiotic family, is one of the most efficient medications used in chemotherapy [8]. Due to its drug resistance and subpar internalization, Dox.H in cancer treatment has been restricted. Dox.H may be delivered by graphene oxide (GO) nanostructures, which can enhance its cytotoxicity when used to treat cancer [9]. To reduce medication resistance and improve the doxorubicin’s cytolytic effect in the fight against cancer, GO NPs are also capable of loading anti-cancer medicines and genetic tools in addition to Dox.H. Theranostic potential has been shown for Dox.H-loaded GO NPs. Although its use as a nanocarrier is restricted by its rough, uneven edges, it is a viable option for drug delivery when integrated into a hydrogel network [10].

To obtain the precise targeting of graphene oxide in tumor cells of the stomach, a dual targeting approach is used; firstly, GO nanoparticles are functionalized with folic acid (Fa) [11]. Folate receptors are considered over-expressed on cancer cell surfaces, enabling them to compete with normal cells for Fa. The amine (NH) group of Fa combines covalently with the carboxylic (COOH) group of graphene oxide [12]. Folic acid’s low immunogenicity, easy availability, and high binding affinity make it more beneficial for modifying GO NPs to target cancer drug delivery [13,14]. Secondly, encapsulation with a bio-compatible, pH-sensitive polymer is used to release the drug-loaded particles at gastric pH in a sustained manner [15].

The development of mucoadhesive-based formulations has long been researched within the gastrointestinal system as a viable site [16]. The use of mucoadhesive polymers to modify the transit duration of delivery systems in a specific region of the gastrointestinal system has attracted the attention of many researchers worldwide [17]. Chitosan (CS), poly (acrylic acid), alginate (SA), poly (methacrylic acid), and sodium carboxymethyl cellulose are some of the several mucoadhesive polymers that have been employed to build oral delivery methods. Achieving mucoadhesion in the digestive system, as a result of the extended residence period and tighter contact between the formulation and the absorptive membrane, may improve the bioavailability of drugs entrapped within a formulation [18]. This could also enable prolonged drug release, lowering the need for repeat administration or lowering the dosage required.

Based on this context, a new gastro-retentive delivery method was developed using nanohybrid SA-CS beads to increase the therapeutic efficacy of Dox.H. Alginate (SA), due to its economic benefits, biocompatibility, biodegradability, particular pH sensitivity, and rapid degradation rates, has undergone substantial research for various biological applications [19]. Additionally, CS is appropriate for biomedical and pharmaceutical formulations because of its biodegradability, nontoxicity, and high biocompatibility. CS has been utilized to enhance or regulate drug release and offer hydrogel beads mucoadhesive characteristics [20].

This study focuses on developing a novel drug delivery system for cancer therapy using graphene oxide (GO) NPs. The conjugation of folic acid to GO introduces a targeted mechanism, enhancing drug delivery precision. The unique hybrid system includes mucoadhesive chitosan, providing a platform for sustained drug release. The addition of graphene oxide in the polymeric system advances the hydrogel’s properties and reduces the chances of an initial burst release of drug. Moreover, the drug is loaded onto the graphene NPs and then encapsulated into pH-sensitive nanohybrid SA-CS beads for potential delivery to the cancerous environment. The primary object of this study is to develop a targeted drug delivery system for cancer therapy, aiming to address the challenges associated with the non-specific delivery of chemotherapeutic agents. The specificity of the drug delivery to cancer cells as a result of conjugating folic acid to graphene oxide is due to the overexpression of folate receptors on the cancer cell surfaces. Moreover, the goal is to fabricate mucoadhesive hydrogel beads using the ion-gelation process, with the intention of improving adherence to the gastrointestinal mucosa for prolonged drug release. We aim to comprehensively characterize the developed hydrogel beads using various techniques, focusing on surface morphology, structural analysis, and homogeneity to ensure the effectiveness of the drug delivery system.

## 2. Materials and Methods

### 2.1. Materials

Graphene oxide (GO) was purchased from Ugent Tech Sdn Bhd, Subang Jaya, Malaysia. Chitosan (CS) with 75–85% deacetylation, a viscosity of 200–800 cP, and a medium molecular weight was obtained from Sigma Aldrich in Darmstadt, Germany. Sodium alginate (SA) with a molecular weight of from 12,000 to 40,000 daltons, chloroacetic acid (ClCH_2_COOH, molecular weight 94.50), ethyl carbodiimide hydrochloride (EDC, molecular weight 155.24), N-Hydroxysuccinimide (NHS, molecular weight 115.09), sodium hydroxide (NaOH, molecular weight 40.00), folic acid (molecular weight 441.40), and anhydrous calcium chloride (CaCl_2_, ≥97%) were also supplied by Sigma Aldrich, Darmstadt, Germany. Doxorubicin HCl with a purity greater than 99% was sourced from Mesochem in Beijing, China. Acetic acid (AA) with 99.5–100.05% purity was purchased from Thermo Fisher Scientific, Waltham, MA, USA. Distilled water (DW) was prepared in the laboratory using distillation equipment.

### 2.2. Preparation of Folic-Acid-Conjugated Graphene Oxide (Fa-GO)

A GO aqueous suspension in 100 mL (1 mg mL^−1^) was bath-sonicated for one hour to yield a clear solution. Following the addition of NaOH (6.0 g) and chloroacetic acid (ClCH_2_COOH) (5.0 g) to the GO suspension, the mixture was bath-sonicated for about 2–3 h to convert the OH groups to COOH via conjugation with acetic acid moieties, resulting in GO-COOH. The resulting GO-COOH solution was neutralized (to a pH of 7.0) with diluted HCl and purified via centrifugation and washing many times. After being dissolved in deionized water, the product was dialyzed for 48 h in a dialysis bag with an MWCO of 8000 against distilled water to eliminate any ions. To increase the stability in physiological media, a sulfonate group was added to GO-COOH. An ary diazonium salt of sulfanilic acid solution was produced as previously described [21]. Sodium nitrite and sulfanilic acid were dissolved in 0.25% NaOH. Dropwise additions of 0.1 N HCl solutions (26 mL) in an ice bath were carried out. The GO-COOH dispersion was placed in a cold bath, and then the diazonium salt solution was added dropwise and stirred for two hours. The SO_3_H-GO-COOH produced was dialyzed for 48 h against distilled water before being stored at 4 °C for future use.

For two hours, 100 mL of a SO_3_H-GO-COOH dispersion was sonicated with NHS (80 mg) and EDC (40 mg). Folic acid was conjugated with GO according to the method previously described by Huang P. et al. [22]. In short, a 0.5% Fa (20 mL) solution was added to the mixture, which was stirred overnight. The end product, Fa-GO, was developed after multiple rinses and filtrations.

### 2.3. Preparation of Nanohybrid SA-CS Beads

In an ultrasonic bath, 20 mg of folic-acid-conjugated GO was dispersed in 50 mL of deionized water. SA was added to the graphene oxide solution and mixed with magnetic stirring for one hour. CS was completely dissolved in a 1% acetic acid solution while being magnetically agitated. Then, CaCl_2_ was added to the freshly prepared 100 mL of CS solution. To fabricate the hybrid beads, 10 mL of SA solution was dripped into 30 mL of CS-CaCl_2_ solution with a syringe needle of 25′ gauge from a distance of 8–10 cm. The beaker containing the bath solution was placed on a magnetic stirrer. After the formation of the hybrid beads, stirring was performed for 15 min at 300 rpm. Following stirring, the bead-containing solution was filtered using a vacuum filtering technique. To rinse the beads, deionized water was used. Finally, the beads were air-dried overnight.

### 2.4. Loading of Doxorubicin HCl (Dox.H)

The drug was initially loaded on Fa-GO to generate the drug-loaded nanohybrid SA-CS beads. Firstly, the Fa-GO dispersion was ultrasonically processed for two hours before the drug was added in a 1:2 *w*/*w* ratio to the Fa-GO dispersion. The dispersion was placed on a magnetic stirrer for 24 h in the dark. After centrifuging the Dox.H supernatant to eliminate unbound particles, the sample was repeatedly rinsed with distilled water. The prepared formulations are given in Table 1.

## 3. Characterization

### 3.1. Percentage Yield

The practical yield of the nanohybrid SA-CS beads was analyzed in order to determine the effectiveness of the production method and to assist in determining which methodology was the most suited. The percentage yield was determined by comparing the weight of the recovered nanohybrid SA-CS beads from each batch to the total weight of the initial components required for the synthesis [23,24]. This was carried out to determine the percentage yield. The following formula was utilized to ascertain the percentage yield:$$\%yield=\frac{Practical\;yield}{Theoretical\;yield}$$

### 3.2. Structural Analysis

The swelling behavior and drug release of synthesized nanohybrid SA-CS beads depend significantly on the morphological characterization of the nanohybrid SA-CS beads [25]. Additionally, the formulation’s structural characteristics may change as a result of interfacial interactions between the GO and the polymer. Therefore, optical and scanning electron microscopes were used to examine the morphological structure of the beads.

#### 3.2.1. Optical Microscope Analysis

By employing an optical microscope with a resolution of 4X and a digital camera with a DCM35 microscope (350K pixels, USB 2.0), the form of the produced beads was examined, and digital photographs were taken.

#### 3.2.2. Field Emission SEM Analysis

Using a scanning electron microscope, the surface morphology of the drug-loaded Fa-GO and nanohybrid SA-CS beads (drug-loaded and unloaded) were examined for structural analysis (Nova Nano SEM model). All the samples were freeze-dried to remove moisture while preserving their structures. The samples were mounted on an SEM stub and then coated with a thin layer of gold to enhance the conductivity and minimize charging during exposure to the electron beam. The coated samples were then examined using a Nova Nano SEM model at magnifications of 1000X and 10,000X with the accelerating voltage set at 5.00 kV to obtain high-resolution images of the surface morphologies [26].

### 3.3. Size Analysis

The stability and particle size distribution of the drug-loaded dispersion and folic-acid-conjugated graphene oxide (Fa-GO) were studied using the dynamic light scattering technique with a Malvern Zetasizer ZS90 (Malvern Instruments, Malvern, UK). A total of 30 µL of the nanoparticle solution was diluted with 2 mL of distilled water to prepare the sample for evaluation. The reported value is the average of three tests conducted on each sample. Moreover, by using a digital micrometer, the size of the manufactured nanohybrid SA-CS beads was meticulously measured (Digimatic Micrometre Mitutoyo, Kanagawa, Japan). Calculations were used to determine each batch’s precise size and weight using the average of 100 nanohybrid SA-CS beads [27]. Each experiment was performed in triplicate. The following equation was used to determine the average size:(1)$$The\;average\;size\;of\;hydrogel\;beads=\frac{the\;sum\;of\;the\;size\;of\;hydrogel\;beads}{total\;no.\;of\;hydrogel\;beads}$$

### 3.4. Homogeneity Analysis

The homogeneity of the developed nanohybrid SA-CS beads was evaluated by weight and size variation analysis [28]. Freshly prepared beads were distributed into five sets and weighed accurately using an electronic balance with a 0.1 mg precision. After weighing, the beads were allowed to dry. The dried beads were weighed again. To determine the weight ratio, the following equation was used:(2)$$Weight\;ratio=\frac{weight\;of\;freshly\;prepared\;beads}{weight\;of\;beads\;after\;drying}$$

### 3.5. Nuclear Magnetic Resonance (H-NMR) Spectroscopy

Nuclear magnetic resonance was utilized to verify that the folic acid and graphene oxide were combined (500 MHZ Bruker). 1H-NMR was performed using a Bruker nuclear magnetic resonance spectroscope. The produced solution was placed in NMR tubes together with graphene oxide, deuterium water, and the conjugate of folic acid and graphene oxide, which was also dissolved in D_2_O. NMR was carried out using a standard shim for DMSO and water.

### 3.6. FTIR Study

For the identification of the chemical species, FTIR of the Fa-GO, SA, CS, Dox.H, and drug-loaded and unloaded nanohybrid SA-CS beads was performed using a Bruker FTIR device. The sample was finely crushed, and the range was selected as between 500 and 4000 with 95% transmittance [29].

### 3.7. Powder X-Ray Diffractometry (PXRD)

The diffraction trends of pure samples of the SA, CS, Fa-GO, drug, and unloaded and drug-loaded nanohybrid SA-CS beads were investigated by powder X-ray diffractometry (D-8 Advance-Bruker, Karlsruhe, Germany) with a Cu target, 40 mA, and 40 kV working conditions. The samples were examined at 2° per min (2θ). The resulting peaks, positions, and shifts were compared with the determined values.

### 3.8. Entrapment Efficiency

Doxorubicin (Dox.H) loading onto the Fa-GO (or GO) was conducted by simply mixing doxorubicin with Fa-GO solution at pH 8 for 24 h in the dark [30]. Unbound excess Dox.H was removed by centrifugation at 12,000 rpm for 5 min and repeated washing with distilled water. The unbound Dox.H was analyzed at a λ-max value of 480 nm using a Lambda 365 UV spectrophotometer (Perkin Elmer, Waltham, MA, USA), referring to the calibration curve prepared under the same conditions. The resulting complexes were re-suspended and stored at 4 °C [31]. The encapsulation efficiency was calculated using the following equation [32,33]:(3)$$Entrapment\;efficiency\left(\%\right)\;=\frac{Total\;amount\;of\;drug-Free\;amount\;of\;drug}{Total\;amount\;of\;drug}\times 100$$

### 3.9. Swelling Study

Unloaded mucoadhesive nanohybrid SA-CS beads at pH 1.2 (the pH of the gastrointestinal environment) and pH 5.6, which is within the range of cancer, were used to assess the swelling trend of the hydrogel beads. A total of 25 mL of the generated buffer at an appropriate pH was held with 10 mg of the dried fabricated nanohybrid SA-CS beads. The hydrogel beads were removed from the solution after a certain time, cleaned with filter paper to remove any extra buffer, and then precisely weighed [34]. The following equation was used to calculate the swelling ratio:(4)$$Swelling\;percentage=\frac{wt\;of\;swollen\;beads-wt\;of\;dry\;beads}{wt\;of\;dry\;beads}\times 100$$

### 3.10. In Vitro Drug Release Analysis

A Type II dissolution apparatus was employed to evaluate the drug release behavior of the prepared nanohybrid SA-CS beads at pH 1.2 and pH 5.6. The buffer simulating gastric fluid at pH 1.2 consisted of hydrochloric acid adjusted to pH 1.2 with 0.2% (*w*/*v*) sodium chloride in distilled water. For the pH 5.6 conditions, the buffer comprised 0.1 M sodium acetate adjusted with acetic acid in distilled water. The release profile of the pure drug, Dox.H, was assessed as a control at pH 5.6. In the dissolution study, 100 mg of the drug-loaded nanohybrid SA-CS beads was carefully enclosed in a dialysis bag and immersed in a basket containing 500 mL of buffer, stirred at 50 rpm, and maintained at 37 ± 0.57 °C. The sink conditions were preserved by replacing the sample buffer (5 mL) with an equal volume at predetermined time intervals (0, 0.5, 1, 2, 3, 4, 6, 8, 10, 12, and every subsequent 2 h up to 24 h). To estimate the amount of drug released, the UV was measured at 480 nm [35]. To determine the average value, readings were taken in triplicate. Moreover, the FTIR analysis was performed for the optimized formulation MNB-6 before and after the drug release experiment at pH 5.6. The comparison was executed to evaluate if any chemical change occurred after the drug was released from the polymeric matrix.

#### In Vitro Drug Release Kinetics

Drug release kinetic models, such as the zero-order, first-order, Higuchi, and Korsmeyer–Peppas models, were used to determine the drug release pattern [36]. Drug release was classified based on kinetic modeling as either non-Fickian (*n* > 0.89) or Fickian (*n* = 0.45) diffusion. The drug release system was regarded as an anomalous system if the obtained value was from 0.45 to 0.89. (non-Fickian).

### 3.11. Ex Vivo Mucoadhesive Analysis

As previously described, in vitro adhesion testing using the wash-off test was used to assess the mucoadhesive properties of the microbeads [37,38]. Fresh goat intestinal mucosa was obtained within 1 h of the animal’s slaughter to maintain tissue integrity and viability. The tissue sample was cleaned by washing it thoroughly with an isotonic saline solution before being mounted on a glass slide measuring 7.5 × 2.5 cm using thread. The prepared slide was securely positioned in the groove of the disintegration test apparatus, assuring stability during the test. Subsequently, 100 beads were meticulously distributed over the moist tissue specimen on the slide.

The apparatus was operated so that the tissue sample was subjected to regular up-and-down movements in a beaker containing 900 mL of phosphate buffer at 37 °C, simulating the natural peristaltic movements of the gastrointestinal tract. At various time intervals up to 3 h, the operation of the apparatus was stopped, and the number of beads still adhering to the tissue was counted. This method provided a controlled environment for assessing the mucoadhesive strength of the hydrogel beads, ensuring consistency and reproducibility in the results. The percentage mucoadhesive strength was calculated by using the following equation:(5)$$Mucoadhesive\;strength\;\%=\frac{No.\;of\;beads\;applied-No.\;of\;beads\;leached\;out}{No.\;of\;beads\;leached\;out}\times 100$$

### 3.12. In Vivo Pharmacokinetic Study

Two groups of healthy rabbits weighing 2.0–2.5 kg underwent an in vivo experiment. The animals were randomly divided into six-rabbit groups, each receiving a dosage of Dox.H equal to 10 mg/kg [39]. Dox.H aqueous solution (1 mg/mL) was used to treat group A, whereas group B received nanohybrid SA-CS beads that were drug-loaded. The Islamia University of Bahawalpur’s Pharmacy Research Ethics Committee authorized the in vivo characterization’s ethical standards. (reference no. of the approved study: PAEC/21/33). The rabbits were housed in separate cages under the same experimental settings and kept under observation for seven days before the in vivo examination. In addition to unrestricted access to water and access to fresh green fodder thrice daily, all animals fasted for 12 h before the experiment. Using a silicon gastric incubation tube, the hydrogel beads were administered to the rabbits. Following dosing, 1 mL of blood was drawn from each rabbit’s jugular vein at various time intervals. A butterfly needle was inserted once, and subsequent blood samples were collected at the stated intervals without additional vein punctures. The blood samples were then placed into tubes containing EDTA as an anticoagulant. To ensure full mixing of the anticoagulant, each blood sample was gently inverted several times. The plasma was separated following a 10 min, 5000 rpm centrifugation and kept at −20 °C. With the aid of 0.4 mL of acetonitrile, the protein in these samples was separated, and 20 µL samples were utilized to calculate the drug concentration using a previously validated HPLC-UV technique [40,41]. The concentration (ng/mL) versus time was used to illustrate the HPLC analytical results (h). The computer-registered program Kinetica 4.1 was used to evaluate the pharmacokinetic parameters [42,43].

### 3.13. Stability Study

An accelerated stability study was used to assess the shelf-life of the optimized formulation of the manufactured nanohybrid SA-CS beads in accordance with ICH recommendations [44]. A stability chamber (Memmert ICH750L, Schwabach, Germany) was used for the stability investigation, which lasted six months while maintaining a constant temperature of 40 ± 2 °C and a humidity of 75 ± 5%. At several points over the six months, the stability was assessed for particle size, percentage drug content, and shape (0, 15, 30, 90, and 180 days). Minitab-17^®^ software was used to determine how long the manufactured nanohybrid SA-CS beads would last.

### 3.14. Statistical Analysis

For each analysis, three triplicates of each experiment were independently performed whilst maintaining the same experimental parameters. The mean value of the obtained results along with the ±standard deviation was given.

## 4. Results

### 4.1. Percentage Yield Analysis

This approach provided valuable information about the production process’s efficiency, enabling well-informed decision-making for future bead synthesis optimization. Assessing the yield can enable researchers to ascertain the feasibility of various production techniques for the development of nanohybrid SA-CS beads. Table 2 presents the percentage yields of several bead batches, ranging from 65.9% to 84.6%. Formulation MNB-7 attained the best yield, likely owing to the optimal ratio of polymers and nanohybrid Fa-GO in its composition.

### 4.2. Optical Microscope

The successful preparation of spherical and smooth-surfaced beads was demonstrated by the optical microscopic images of the fabricated nanohybrid SA-CS beads, as illustrated in Figure 1. Figure 1a shows a digital photograph of the freshly prepared unloaded MNB-6 sample. No morphological differences were seen between the different prepared unloaded formulations of the nanohybrid SA-CS beads. In the image, the drug-loaded hydrogel beads represent the inner portion of the SA-CS beads, shown by a thinner area with an immensely dense outer (dark black) area. The structure of the beads became chiral toward the center as the quantity of GO in the polymer solution increased, as shown in Figure 1b. The optical images are inconsistent with the implementations Chenlu Bao et al. reported in their study on using GO hydrogel beads to rapidly clean dangerous substances [45]. Furthermore, after being dried, the hydrogel beads changed in form and size as the interior water was lost and the polymeric network partially collapsed [46].

### 4.3. Scanning Electron Microscope (SEM) Analysis

Figure 2a represents the SEM image of the drug-loaded Fa-GO sample. The layered structure revealed a smooth and layered surface, indicating that the drug was appropriately loaded onto the Fa-GO conjugate; however, a few uneven edges can be seen in the image seen due to the conjugation of the folic acid. The GO surface was observed to have a lamellar-like structure with extremely thin, homogeneous graphene layers that seemed wrinkled [47]. Similar results were obtained by Deb A et al. during their study of functionalized graphene for dual drug delivery [48]. Figure 2b,c shows the morphological structure of the nanohybrid SA-CS beads with different concentrations of Fa-GO. There was little difference between the morphological appearances of the MNB-1 and MNB-6 samples. The wrinkles indicate the crosslinking between the SA-CS polymer matrix and Fa-GO. As the concentration of Fa-GO increased, the SEM images showed a more compact surface. The results obtained are inconsistent with the SEM images captured by Javanbakht S et al. during their SEM investigation of graphene-quantum-dot-encapsulated hydrogel beads [49]. Figure 2d represents the drug-loaded MNB-6 sample; the surface displayed a rough surface compared to the unloaded hydrogel beads. The roughness of the nanohybrid SA-CS beads was related to the drug molecules entrapped in the polymer matrix [50].

### 4.4. Size Analysis

The toxicity of graphene oxide (GO) is dependent on the size of its sheets, with smaller sheets exhibiting lower toxicity. This characteristic is crucial for the use of GO in therapeutic applications [51]. The size and heterogeneity of the folic-acid-conjugated graphene oxide (Fa-GO) and drug-loaded Fa-GO were assessed using a zeta sizer following ultrasonication. The zeta sizer measurements revealed that the size of the folic-acid-conjugated graphene oxide (Fa-GO) was around 554.2 ± 95.14 nm. However, when the drug was loaded onto the graphene oxide π-π conjugate, the size increased to 778.6 ± 186.7 nm, as indicated in Table 3 and Figure 3. This increase in size corresponded to the successful loading of doxorubicin onto the Fa-GO NPs. The preceding literature likewise presents similar results [52].

The zeta potential of the Fa-GO was measured at the same concentration, yielding a value of −65 ± 12.33 mV. This indicates that the surfaces of the Fa-GO samples were negatively charged due to the presence of hydrophilic carboxyl groups. The results demonstrate that the drug-loaded Fa-GO had a lower value of −69 ± 23.45 mV compared to the unloaded Fa-GO, which can be attributed to the considerable disparity in the surface charge. Particles in suspension with a significant negative or positive zeta potential will not combine because they have a natural inclination to repel each other [53]. Consequently, the solubility of the Fa-GO in water was enhanced through the process of drug loading. The previous literature [26] has also demonstrated an increased zeta potential and stability of graphene oxide coupled with folic acid. The alterations in the thickness, roughness, and zeta potential demonstrate the effective binding of the Fa with GO [54].

The average size of the hydrogel beads generated from each group closely matched the values reported in Table 4. However, it was discovered that increasing the amount of Fa-GO resulted in a reduction in the size of the beads. This size reduction may be attributed to the formation of robust hydrogen bonds between the polymer chains and the GO. Furthermore, the nanohybrid SA-CS beads exhibited a reduction in size when the concentration of CS increased. The increased contact resulted in a denser and undamaged structure of the nanohybrid SA-CS beads [55].

### 4.5. Homogenous Nature

Figure 4 displays the results of the investigation of the weight variations. Every formulation resulted in a linear horizontal plot, suggesting that the ratio of wet to dry beads stayed the same across all formulations sharing the same formulation. These minimal variations in the computed weight ratio of hydrogel beads from the same lot indicate the sample’s exceptional homogeneity. It follows that the uniformity of the hydrogel beads was achieved by maintaining constant experimental conditions during their preparation. According to Bajpai et al., beads of a homogeneous nature were produced during a calcium release investigation using sodium alginate–dextran hybrid hydrogel beads when the experimental conditions were kept consistent [28].

### 4.6. NMR Spectroscopy

The conjugation of the folic acid with the graphene was examined by 1H-NMR, as presented in Figure 5. The individual spectra of the folic acid were taken using deuterium dimethyl sulfoxide as a solvent, and the spectra of the GO and Fa-GO conjugate were taken using D_2_O as a solvent. There are no apparent resonance peaks in the GO 1H NMR spectra between 0 and 4 ppm. D_2_O was given credit for the peak at 4.7 ppm. Almost comparable results were stated by Kavitha T. et al. during their NMR study to confirm the polymeric functionalization of GO [56]. However, the spectra of folic acid showed results at from 6.735 to 8.484 ppm that corresponded to the aromatic ring of folic acid [57]. The folic acid–graphene oxide complex spectra clearly depicted peaks at 4.4, 6.707, and 8.484 ppm, confirming the successful conjugation.

### 4.7. FTIR Analysis

Figure 6a displays the FTIR spectrum of pure CS, which had distinctive peaks at 2932 cm^−1^ and 2862 cm^−1^, demonstrating the stretching of both symmetric and asymmetric -CH. Amide I was denoted by the presence of C=O stretching at 1652 cm^−1^, amide II was denoted by the presence of N-H bending at 1570 cm^−1^, and amide III was denoted by the presence of C-N stretching at 1379 cm^−1^. Peaks between 1400 and 1370 cm^−1^ indicate CH_2_ bending and CH_3_ stretching, while the absorption bands at 1070 and 1024 cm^−1^ serve as indicators of C-O stretching. The SA polymer’s FTIR spectrum is depicted in Figure 6b. The peak at 1465 cm^−1^ indicates the average bending of CH, which belonged to the alkane methylene group, whereas C-O-O exhibited intense wide stretching at around 1050 cm^−1^ [58]. Figure 6c shows the FTIR spectrum of the model drug Dox.H; the peaks between 764 and 871 denote the wagging of the primary amine group and the deformation of the N-H bond. Near 1114 cm^−1^, the C-O-CH_3_ band was shown to be stretching [59]. The FTIR spectrum of pure Fa is shown in Figure 6d at 1607 cm^−1^ due to the N-H bending vibration of the CONH group, whereas the spectrum at 1735 cm^−1^ shows the C=O amide stretching of the carboxyl group. The spectra at 1463 cm^−1^ demonstrate the absorption band of the phenyl ring [21].

Figure 6e shows the FTIR spectra of Fa-GO, showing a characteristic peak at 3400 cm^−1^ that indicates the presence of an O-H bond, a peak at 1734 cm^−1^ that denotes the presence of a C=O bond, a peak at 1632 cm^−1^ that denotes the presence of a C=C bond, and a peak at 1043 cm^−1^ that denotes the presence of C-O. The presence of a peak at 1643 cm^−1^ and an additional aromatic C-H bending peak at 860 cm^−1^ both confirm the conjugation of folic acid in the Fa-GO, respectively, while the absorption peaks seen between 1000 and 1200 cm^−1^ can be attributed to bands arising from the vibration modes of the sulfonic acid groups used during the functionalization to increase the stability of the GO in the physiological media [60,61]. The FTIR spectrum of the drug-loaded Fa-GO at the characteristic peak of folic acid of 1647 cm^−1^ is shown in Figure 6f. The peak of the -OH group is shown at 3417 cm^−1^ with a little shift to the lower band. The spectra at 2934 cm^−1^ indicate the stretching of the C-H band. The main amine NH2 wagging and N-H deformation bonds were responsible for the maxima at 854 and 760 cm^−1^ [62]. The extra absorbance bands in the spectrum demonstrate that the Dox.H was effectively loaded onto the Fa-GO sample. Figure 6g displays the nanohybrid SA-CS beads’ FTIR spectrum. It shows all the same characteristic peaks of SA, CS, Fa, GO, and the drug, showing that the nanohybrid SA-CS beads were successfully made. The interactions between the Fa-GO and alginate and the Fa-GO with chitosan and alginate were supported by the shifts and overlaps in these characteristic peaks, confirming the formation of a stable polyelectrolyte complex with controlled drug release properties.

### 4.8. PXRD Analysis

The PXRD pattern of Dox.H, shown in Figure 7a, demonstrates the drug’s highly crystalline nature, as various firm diffraction peaks were obtained at 13.02, 17.56, 22.48, 27.02, 32.33, and 39.11° at 2θ. The peaks were in the range values investigated by previous studies [63]. The crystalline nature of the CS was indicated by peaks at 11.07 and 31.22°, as shown in Figure 7b [64]. The PXRD pattern of the pure SA (Figure 7c) gave the relatively sharp peak at 28.09° at 2θ, which provided a d-spacing of about 3.17 Å, as calculated by the Bragg equation. These results indicate the amorphous nature of the polymer and lie in a similar range to previous studies [65]. The PXRD pattern of the Fa-GO displayed broader and less defined peaks (Figure 7d), suggesting a less organized and more disordered structure of the folic-acid-functionalized graphene oxide. The amorphous nature of materials such as graphene oxide is characterized by the presence of functional groups that disturb the normal arrangement of graphene layers, resulting in diffraction peaks that are wide in range. However, when the drug was loaded on the Fa-GO conjugate, additional peaks at 11.09, 26.43, and 28.97° were seen, as shown in Figure 7e. The presence of these additional peaks indicates the successful integration of the Dox. H into the Fa-GO structure. Furthermore, these peaks suggest that the drug molecules brought about a distinct crystalline arrangement inside the composite material. The presence of these distinct peaks indicates the development of a more structured crystalline phase, which can be ascribed to the organized alignment of the doxorubicin molecules on the Fa-GO surface. These PXRD pattern results were compared to the graphene-like structure in the JCPDS 01-0646 database [66]. The interaction between the Dox.H and Fa-GO was facilitated by π-π stacking and electrostatic interactions, which enhanced the arrangement of the Dox.H molecules in a more orderly manner. The packing arrangement resulted in a greater level of crystallinity in the Dox.H-loaded Fa-GO composite in comparison to the comparatively amorphous Fa-GO alone. Therefore, the appearance of these additional diffraction peaks in the PXRD pattern provides evidence of the successful introduction of the Dox.H and emphasizes the alterations in the structure caused by the inclusion of the Dox.H, resulting in a material with a unique crystalline arrangement.

Figure 7f represents the PXRD results of the fabricated drug-loaded nanohybrid SA-CS beads. The results showed no sharp characteristic peaks, indicating the presence of a strong interaction between the polymeric matric and the functionalized NPs. After the hydrogel matrix system was formed, the diffraction peak of Fa-GO vanished, confirming the homogenous distribution of the Fa-GO into the polymer gel. No characteristic drug or Fa-GO peaks were seen in the drug-loaded nanohybrid SA-CS beads. Previous studies have also presented similar results, where diffraction peaks were minimized after forming aerogels, specifying that the GO sheets were well exfoliated within the polymer matrix [67,68]. This amorphous nature of the fabricated drug-loaded mucoadhesive hydrogel beads is well known in drug delivery applications, as it results in them being more effective than crystalline compounds due to their effective interaction with the biological system [69].

### 4.9. Encapsulation Efficacy

Drug loading and release behavior are the most important characteristics to consider when evaluating a drug delivery system. The sp^2^-hybridized and conjugated structure of GO can have π-π stacking interactions with the quinine portion of doxorubicin, leading to effective loading. In addition, the amine (-NH_2_) and hydroxyl (-OH) groups on doxorubicin can induce a strong hydrogen bonding interaction with the -COOH and -OH groups on Fa-GO. The loading of Dox.H onto Fa-GO can also be attributed to simple π-π stacking and hydrophobic interactions, according to previous studies. The drug loading efficiency was found to be 72.07%, as shown in Table 4. These values are higher than the values previously reported for folic-acid-conjugated carbon nanotubes [70].

Altering the concentrations of the polymer and Fa-GO allowed for determining which formulation delivered the most efficient encapsulation and, hence, the highest drug loading capacity. It was demonstrated that the entrapment efficiency varied with the polymer and folic-acid-conjugated graphene oxide concentrations. The created nanohybrid SA-CS beads were made more resistant to degradation by adding additional polymer. Large amounts of alginate provide appropriate encapsulation, which lowers the possibility of drug leakage and increases trapping [71]. The nanocomposite beads exhibited great entrapment efficiency as the CS content rose. The CS gave the beads excellent structural stability, which led to the creation of an interpenetrating polymer matrix [72]. The findings of the investigation into how the Fa-GO concentration affected the entrapment of the manufactured nanocomposite hydrogel beads are shown in Table 5. It was demonstrated that as the number of nanohybrids increased, so did the amount of Dox.H present in the polymeric network. By generating larger interstitial gaps and pores in the polymeric alginate matrix, the Fa-GO NPs in the nanohybrid SA-CS beads delivered a high entrapment efficiency. Graphene oxide and Dox.H have a strong electrostatic attraction and form hydrogen bonds, both of which contributed to the compound’s high loading percentage. Therefore, it is inferred that the entrapment effectiveness of the nanohybrid SA-CS beads could be improved by increasing the amount of Fa-GO used. The encapsulation effectiveness of the MNB-6 sample increased to 83.25%. Similar improvements in terms of the encapsulation effectiveness of nanocomposite hydrogel beads were also reported by Rasoulzadeh M. et al. [55].

### 4.10. Swelling Study

In vitro swelling studies are crucial because they describe how drug delivery systems release their contents in various media. To identify the behavior of the fabricated nanohybrid SA-CS beads in different physiological conditions of the stomach, a swelling study was performed at pH 1.2 (normal pH of the gastric environment) and pH 5.6 (pH range of gastric cancer) [73]. At pH 1.2, the fabricated nanohybrid SA-CS beads first showed rapid swelling followed by slow swelling until the size reached a constant value; the fast swelling of the nanohybrid SA-CS beads occurred due to hydration and osmotic pressure, which led to the migration of water molecules into the hydrogel beads [71]. Moreover, the CS on the surface of the hydrogel matrix also imparted an increment in swelling due to its pH-sensitive behavior. In addition, the second phase of the polymeric matrix recoiling was caused by the release of calcium ions from the alginate gel matrix’s egg-box cavity, which un-ionized the carboxyl groups to produce insoluble alginic acid and cause the water to drain and the hydrogel matrix to coil. Even though CS is highly soluble in an acidic environment and is charged for converting its amine units into an NH3+ soluble state, the interaction between the amino groups and protonated carboxylic groups was insufficient to cause swelling. As a result, the calcium alginate structure was dominant in the limited overall swelling behavior. Additionally, as the concentration of graphene oxide increased, the swelling ratio was decreased due to the forming of a more compact structure of the hydrogel matrix. The results correlate with previous studies demonstrating a similar behavior of CS and SA [74]. At pH 5.6, the nanohybrid SA-CS beads presented a relatively high but stable swelling pattern, as shown in Figure 8b. Osmotic-pressure-driven water molecule migration within the beads caused the initial increase in swelling. The alginate concentration had a direct impact on how much the hydrogel beads expanded. The dissolution of the COO was caused by the expansion of polymeric chains. The egg-box structure of the calcium alginate was opened due to the chelating effect of phosphate ions [75]. However, in the case of the SA-CS polymeric complex, the hydrogel matrix’s stability may have been a result of adding the CS into the dense matrix; the CS’s ability to cause swelling increased along with its concentration. Likewise, the folic-acid-conjugated graphene oxide enhanced the swelling of the hydrogel beads, with time showed no degradation, and maintained the shape. This swelling behavior was due to physical crosslinking between the polysaccharide macromolecules and graphene oxide [34]. The studied nanohybrid SA-CS beads may be an appropriate carrier for drug delivery in an acidic pH cancerous environment, according to the observed swelling pattern.

### 4.11. Drug Release Study

The drug release pattern at pH 1.2 (as shown in Figure 9) exhibited a more regulated pattern compared to at pH 5.6. Due to the un-ionization of the carboxylic group, there was a reduction in the size of the polymeric network in the acidic pH environment, resulting in a reduced rate of drug release. Increasing the amount of graphene oxide in the hydrogel beads led to a decrease in the release of the drug. This was because the pores became smaller, and more carboxylic groups were included in the graphene oxide [76]. In addition, the drug release at pH 1.2 may be attributed to the hydration and swelling of the beads, which enabled the diffusion of the drug molecules.

Figure 9 shows the drug’s in vitro release from the nanohybrid SA-CS beads. This study’s main goal was to guarantee a more controlled and prolonged drug release pattern from the hybrid polymeric matrix while avoiding the issues with the non-hybrid polymeric technology. One of these issues is hydrogel matrix erosion, which causes drug particles to discharge abruptly [77,78]. To prevent erosion and drug release from the hydrogel matrix’s surface, the drug was first loaded onto the folic-acid-conjugated graphene oxide by “π-π stacking” and then wrapped in a polymeric matrix of SA and CS. pH 5.6 was used to conduct the drug release analysis (endosomal pH of cancer cells). The nanohybrid SA-CS beads produced an extended and regulated release trend at pH 5.6. The amount of GO, SA, and CS determined how rapidly the drug was released. As the concentration of GO and the polymer increased, the drug release curve became steadier and more stable because they acted as dual drug-binding factors. Initially, the drug release was high, and then it attained a steady equilibrium, so the nanohybrid SA-CS beads showed a biphasic drug release pattern. The hydrogen bonding in the polyelectrolyte complex initially broke down, causing the complex to dissociate. This was followed by the ionization of the polymer chains, which had been physically entangled to form the complex. The polymers’ electrostatic attraction to one another uncoiled the chains. The result was that the beads became swollen. As a result, the drug diffused out of the matrix as the beads swelled. The nanohybrid SA-CS beads were used to observe the progressive rise in the percentage release. The MNB-6 sample provided the maximum drug release of 68.29% at pH 5.6 during a 24 h investigation. The effectiveness of the drug encapsulation and hydrogel bead swelling were closely correlated with the drug release; the hydrophilicity of the drug carriers was imparted by the hydroxyl and carboxyl groups of the hydrogel matrix. Additionally, according to Zahra Mirzaie et al., the inclusion of graphene oxide made hydrogels more compact and controlled the release of the drug [79].

The FTIR analysis was performed in combination with a dissolution study to identify the chemical modification after the drug was released from the nanohybrid SA-CS beads. The results indicate that the characteristic peaks of the drug at 764 and 871 disappeared after its release, whereas the sharp band indicating the presence of C-O-CH_3_ was also minimized, indicating the drug’s release from the polymeric matrix [59]. However, no major change was observed in the chemical structure of the hybrid polymeric matrix as the drug was physically loaded through π-π stacking onto the Fa-GO NPs, as shown in Figure 9.

#### Drug Release Kinetics

The obtained values of the percentage drug release were fitted into different kinetic models, including the zero-order, first-order, Higuchi, and Korsmeyer–Peppas models in order to better represent the release mechanism of the drug-loaded formulations. According to the results, the closest regression coefficient value was 1. The most appropriate model for that formulation was the one that displayed the closest regression coefficient (R) value. The calculated values are listed in Table 6. All nanohybrid SA-CS beads displayed first-order kinetics, predicated on the hypothesis that the amount of drug released from the formulation relies on the amount of entrapped drug. The sort of swelling pattern demonstrated by the formulation was identified by the Korsmeyer–Peppas model. Given that the resulting value was smaller than 0.45, the derived formulations demonstrated a Fickian release pattern. Fickian diffusion is indicated if the value of *n* is equal to or less than 0.45. When the value of *n* is between 0.45 and 0.89, it is non-Fickian.

Additionally, the increase in the concentration of the Fa-GO and polymer improved the matrix’s compact structure and caused the non-Fickian dynamics to switch to Fickian kinetics. These findings are consistent with earlier findings by Meng Xun et al. [80].

### 4.12. Ex Vivo Mucoadhesive Analysis

In multi particulate systems, mucoadhesion is a significant property because it can prevent the quick clearance of particles due to peristalsis and prolong contact with the target location [81]. Wash-off tests utilizing sheep intestinal mucosa were used to measure the mucoadhesive strength. Increases in the CS concentration increased the mucoadhesion percentage, but increases in the Fa-GO concentration resulted in a decrease in the mucoadhesion percentage. The improved mucoadhesion outcomes of the SA-CS hydrogel beads can be explained by the electrostatic interactions between the positive charge of the CS and the negative charge of the glycoprotein on the mucosal surface [82]. Therefore, adding CS to the Fa-GO hybrid SA matrix enhanced the mucoadhesion of the developed beads and minimized their leaching, as shown in Figure 10.

### 4.13. In Vivo Pharmacokinetics

The ideal mobile phase contained acetonitrile and water at a ratio of 32:68 (*v*/*v*), with the pH being raised to 2.6 by 85% orthophosphoric acid. HPLC was used to analyze a 20 µL sample introduced into the system. The UV detector was set at 233 nm, and the mobile phase was run at a flow rate of 1 mL per minute. A C18 column (4.6 mm 150 mm, 5 μm) was employed, with the HPLC column temperature controller set at 35 °C. Table 7 lists the values of several pharmacokinetic parameters. Figure 11 shows the construction of the plasma concentration of Dox.H against the time interval following the administration of the drug solution to the control group and the treatment of the experimental groups with the fabricated nanohybrid SA-CS beads. According to the results, the drug was absorbed more efficiently and reached its maximum peak within eight hours when administered as an oral solution. Group B was given a treatment with the nanohybrid SA-CS beads, which offered better-controlled release of the hydrogel beads than the oral solution of Dox.H. This treatment resulted in plasma concentration after twenty-eight hours. However, the Cmax of the fabricated mucoadhesive beads was 2.58-fold greater than that of the drug solution. Similarly, the AUC 0–∞ of the nanohybrid SA-CS beads was higher at 7428.1 μg/mLh, which was 3.56 times greater than the value obtained for the drug solution. The calculated plasma concentration revealed that the prepared nanohybrid SA-CS beads were ideal for the sustained release of the drug for a relatively extended period. These findings are supported by the research of D. R. Kalaria et al., who developed NPs for the oral administration of doxorubicin [39].

### 4.14. Stability Study

The optimized formulation MNB-6 (high entrapment efficiency) was subjected to an accelerated stability study. According to ICH guidelines, the stability study was conducted to evaluate the potential of the fabricated formulation to withstand variable environmental conditions. No significant difference (*p* > 0.05) was observed in terms of the particle size and drug content before and after the study, as shown in Table 8. The morphology of the tested formulation appeared to be the same even after six months of accelerated study. The shelf-life was calculated by the percentage of the nanohybrid SA-CS beads’ drug content reaching 90%. The determined rate constant and shelf-life were 8 × 10^−5^ (day^−1^) and 1312.5 days, respectively. This revealed that the developed nanohybrid SA-CS beads were shelf-stable for more than two years, according to the above standards. The results correlated with those of a study regarding microbeads of locust bean gum and sodium alginate prepared by Upadhyay et al. [83].

## 5. Conclusions

The dripping extrusion process was used to successfully create nanohybrid SA-CS beads for gastro-retentive drug delivery. Due to the uniformity of the parameters, all of the produced hybrid formulations were homogenous. Additionally, a polymeric matrix was able to enclose the rough edges of the GO. The nanohybrid SA-CS beads that were developed included CS, which gave them a mucoadhesive quality. Optical and scanning electron microscopy studies indicated the wrinkled surface morphology of the nanohybrid SA-CS beads, in which a polymeric matrix was coating the craggy edges of the GO. The results from the FTIR and XRD analyses confirmed that the hydrogel matrix was effective in encapsulating the of Fa-GO and the successful loading of the drug. Ex vivo mucoadhesive studies showed that higher concentrations of CS improved the mucoadhesive properties. At pH5.6, the nanohybrid SA-CS beads showed promising swelling characteristics and drug release. The hydrogel beads had a shelf-life of over two years, according to an ICH-guideline-based accelerated stability study. Based on the findings, it is suggested that the nanohybrid SA-CS beads generated could be considered as a promising option for gastro-retentive drug delivery.

The limitations of this study include its reliance on in vitro assays to assess the drug release and mucoadhesive properties, which may not accurately represent actual in vivo performance in a biological setting. Although the stability study adhered to ICH guidelines, it did not evaluate long-term storage conditions, potentially impacting the practical applicability of the hydrogel beads’ shelf-life. Additionally, incorporating thermal stability testing and conducting cell line studies could provide further insights into the performance of the nanohybrid SA-CS beads, which we plan to address in our future project.

## Figures and Tables

**Figure 1 pharmaceutics-16-01451-f001:**
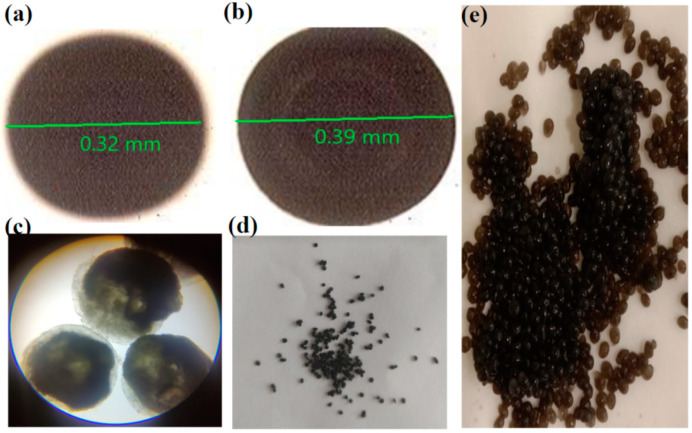
Optical image of (**a**) unloaded, (**b**) drug-loaded, and (**c**) dry nanohybrid SA-CS beads, and digital images of (**d**) dry hydrogel beads and (**e**) freshly prepared nanohybrid SA-CS beads.

**Figure 2 pharmaceutics-16-01451-f002:**
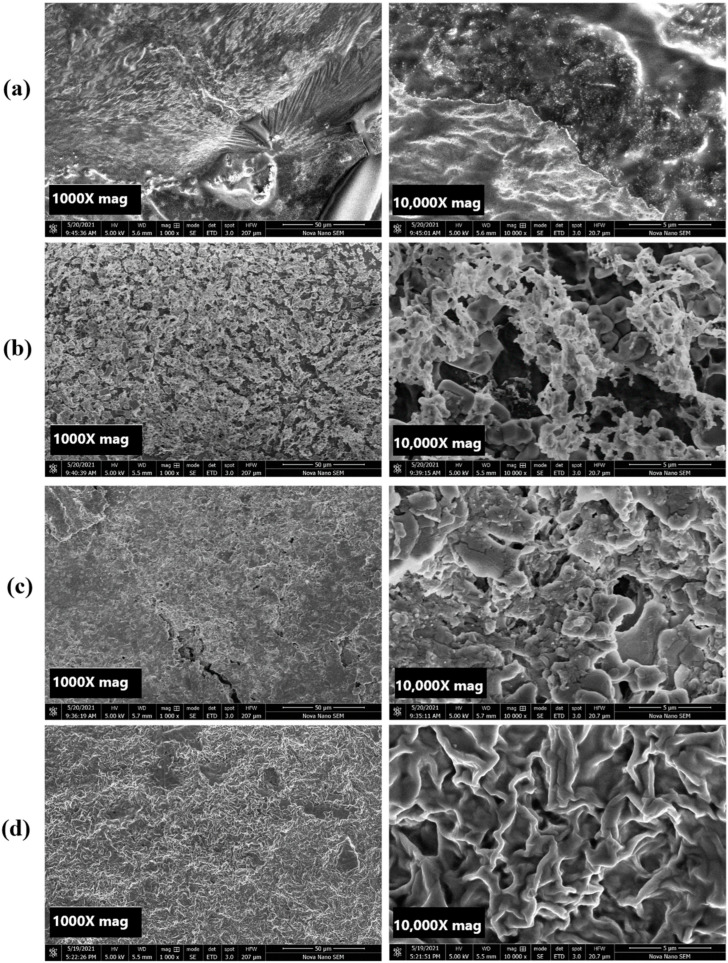
SEM image at 1000X and 10,000X magnification of (**a**) folic-acid-conjugated graphene oxide (Fa-GO), (**b**) MNB-1 sample, (**c**) MNB-6 sample, and (**d**) drug-loaded MNB-6 sample.

**Figure 3 pharmaceutics-16-01451-f003:**
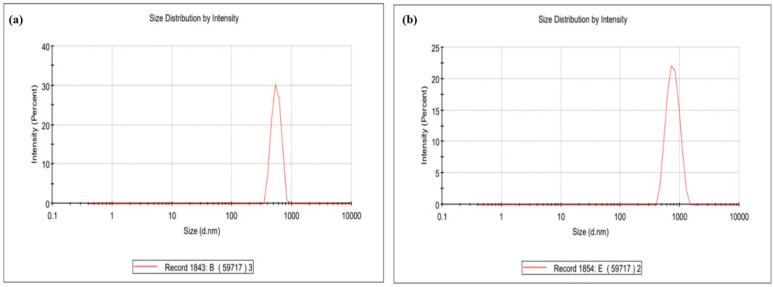
Graphical representation of zeta potential, size, and PDI of (**a**) folic-acid-conjugated GO (Fa-GO) and (**b**) doxorubicin-loaded Fa-GO.

**Figure 4 pharmaceutics-16-01451-f004:**
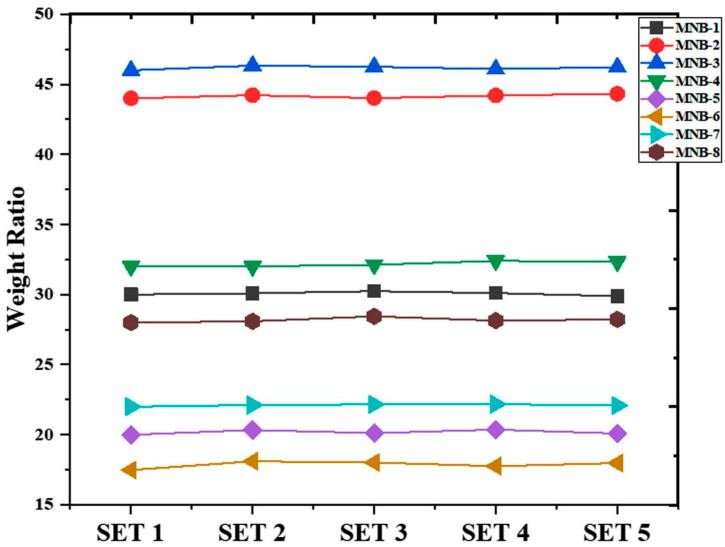
Weight variation analysis of prepared nanohybrid SA-CS beads.

**Figure 5 pharmaceutics-16-01451-f005:**
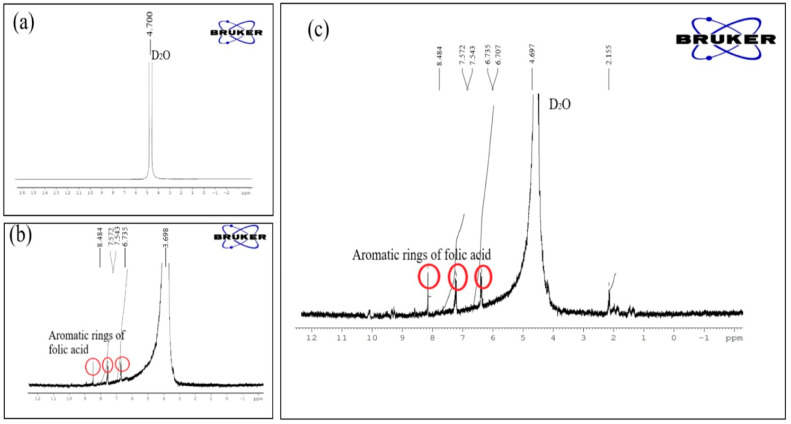
NMR spectra of (**a**) graphene oxide (GO), (**b**) folic acid (Fa), and (**c**) folic-acid-conjugated graphene oxide (Fa-GO).

**Figure 6 pharmaceutics-16-01451-f006:**
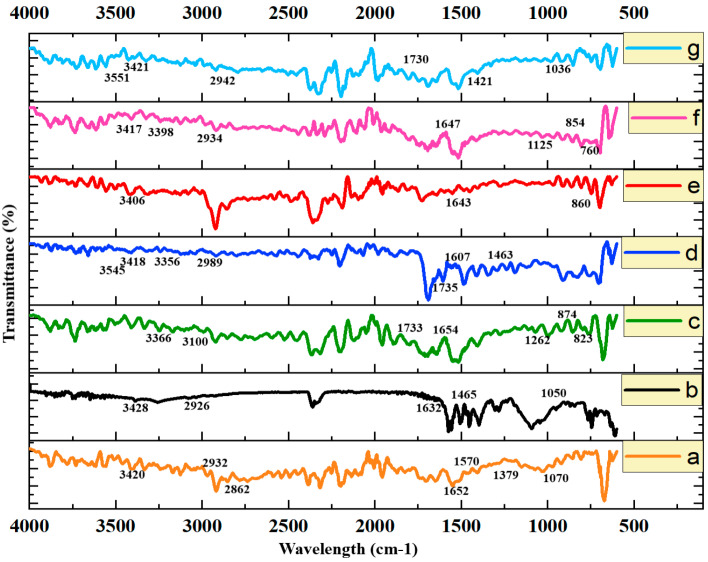
Cumulative FTIR spectra of (**a**) pure chitosan, (**b**) pure sodium alginate, (**c**) model drug, (**d**) pure folic acid, (**e**) Fa-GO, (**f**) drug-loaded Fa-GO, and (**g**) nanohybrid SA-CS beads.

**Figure 7 pharmaceutics-16-01451-f007:**
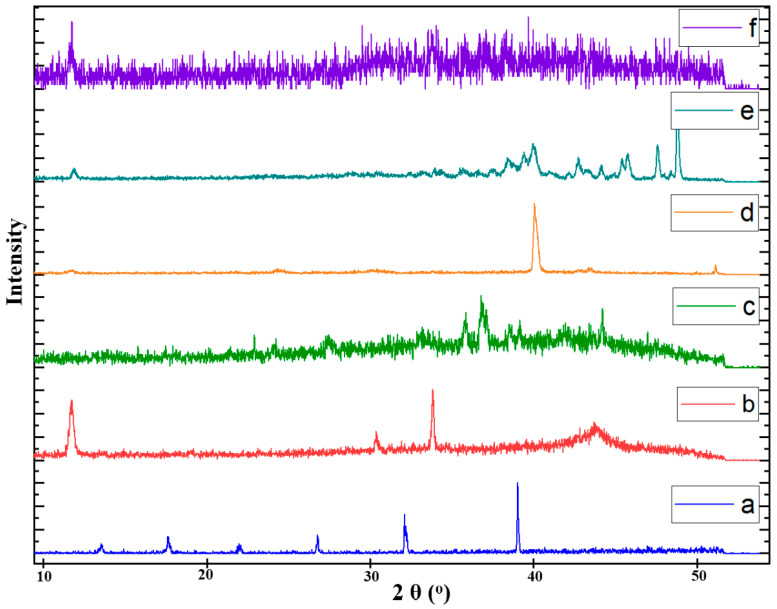
PXRD pattern of (**a**) drug (doxorubicin), (**b**) pure chitosan, (**c**) pure sodium alginate, (**d**) Fa-GO, (**e**) drug-loaded Fa-GO, and (**f**) fabricated drug-loaded nanohybrid SA-CS beads.

**Figure 8 pharmaceutics-16-01451-f008:**
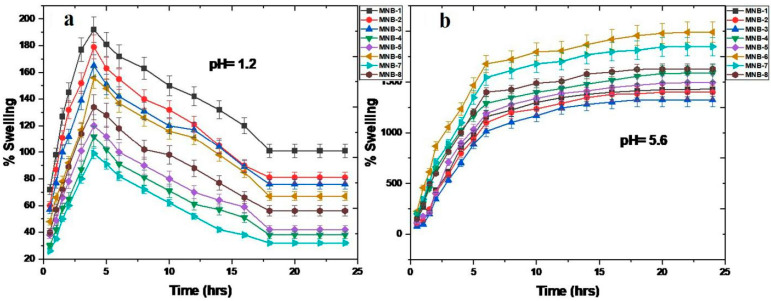
(**a**) Percentage swelling of fabricated nanohybrid SA-CS beads without drug at pH 1.2. (**b**) Percentage swelling of fabricated nanohybrid SA-CS beads without drug at pH 5.6.

**Figure 9 pharmaceutics-16-01451-f009:**
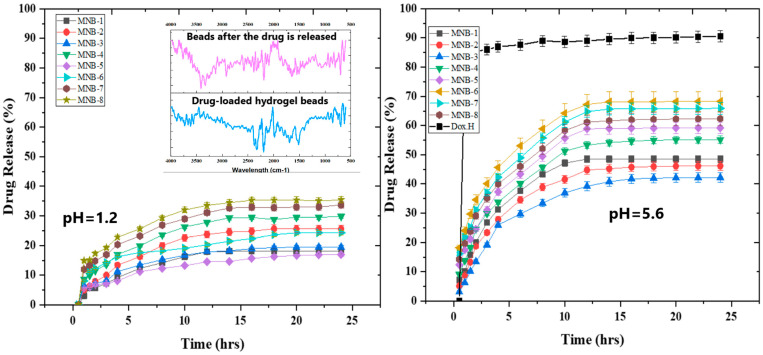
Percentage drug release study of doxorubicin-loaded nanohybrid SA-CS beads. Comparison of FTIR spectra of hydrogel beads before and after drug release.

**Figure 10 pharmaceutics-16-01451-f010:**
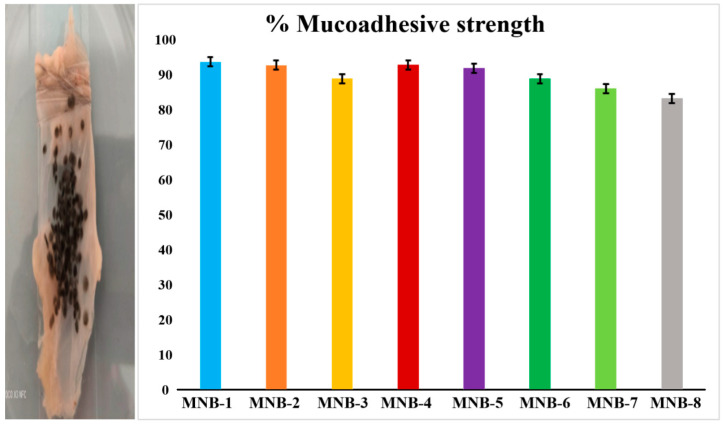
Mucoadhesive analysis of fabricated nanohybrid SA-CS beads.

**Figure 11 pharmaceutics-16-01451-f011:**
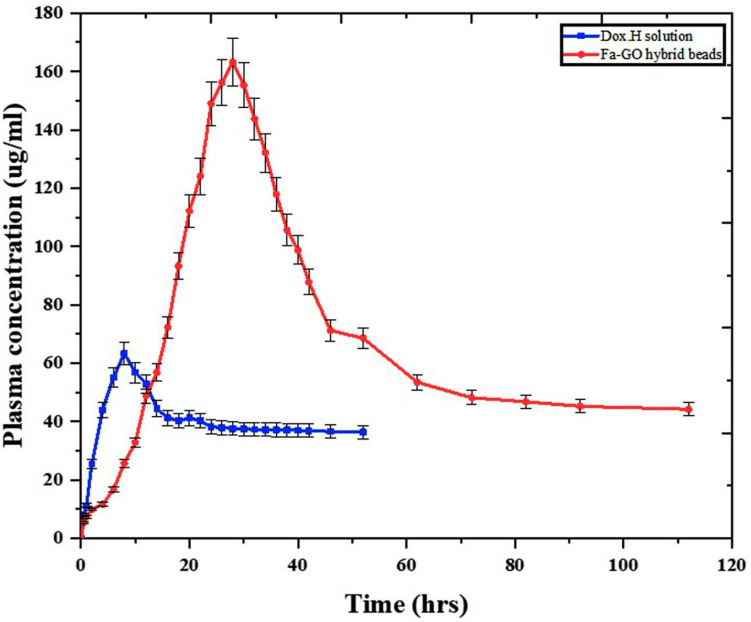
Doxorubicin HCl (drug) in vivo plasma concentration vs. time administered orally as doxorubicin solution and doxorubicin-loaded fabricated nanohybrid SA-CS beads. Data represent mean ± S.D. (*n*  =  3).

**Table 1 pharmaceutics-16-01451-t001:** Various developed formulations of nanohybrid SA-CS beads.

Formulations	SA (g/100 mL)	Fa-GO (mg/100 mL)	CS (g/mL)	CaCl_2_ (g/100 mL)
MNB-1	3.5	2.5	0.3	5
MNB-2	3.5	2.5	0.2	5
MNB-3	3.5	2.5	0.1	5
MNB-4	3.5	5	0.3	5
MNB-5	3.5	5	0.2	5
MNB-6	3.5	5	0.1	5
MNB-7	4.0	5	0.1	5
MNB-8	3.0	5	0.1	5

Note: Sodium alginate (SA), chitosan (CS), calcium chloride (CaCl_2_), and folic-acid-conjugated graphene oxide (Fa-GO).

**Table 2 pharmaceutics-16-01451-t002:** Percentage yield of fabricated Fa-GO nanohybrid hydrogel beads.

Formulation	Percentage Yield
MNB-1	68.8± 1.54
MNB-2	67.2 ± 1.33
MNB-3	65.9 ± 2.31
MNB-4	82.8 ± 2.09
MNB-5	82.2 ± 1.16
MNB-6	81.6 ± 1.72
MNB-7	84.6 ± 1.48
MNB-8	77.3 ± 2.05

**Table 3 pharmaceutics-16-01451-t003:** Zeta potential, size, and PDI of folic-acid-conjugated GO (Fa-GO) and doxorubicin-loaded Fa-GO.

Formulations	Zeta Potential (mV)	Size (nm)	PDI
Fa-GO	−65 ± 12.33	554.2 ± 95.14	0.637
Dox.H-loaded Fa-GO	−69 ± 23.45	778.6 ± 186.7	0.361

**Table 4 pharmaceutics-16-01451-t004:** Particle size and the weight ratio of fabricated Fa-GO hybrid alginate–chitosan hydrogel beads.

Formulation	Particle Size (mm)	ANOVA *p*-Value (Particle Size)	Average Weight Ratio	ANOVA *p*-Value (Weight Ratio)
MNB-1	0.37 ± 0.07	0.02	30.45 ± 1.02	0.01
MNB-2	0.42 ± 0.03		44.65 ± 2.19	
MNB-3	0.45 ± 0.12		47.29 ± 1.16	
MNB-4	0.38 ± 0.04		32.65 ± 0.98	
MNB-5	0.34 ± 0.11		21.65 ± 1.17	
MNB-6	0.32 ± 0.06		18.02 ± 1.53	
MNB-7	0.35 ± 0.09		22.17 ± 0.78	
MNB-8	0.37 ± 0.08		28.67 ± 2.43	

Note: All values are expressed as mean ± SD (*n* = 3). A value of “*p*” less than 0.05 was considered significant.

**Table 5 pharmaceutics-16-01451-t005:** Encapsulation efficiency of Dox.H-loaded folic-acid-conjugated graphene oxide (Fa-GO) and fabricated hydrogel beads.

Formulation	Encapsulation Efficiency (%)
Fa-GO	72.07 ± 1.62
MNB-1	64.23 ± 1.77
MNB-2	61.23 ± 2.17
MNB-3	57.56 ± 1.11
MNB-4	73.89 ± 1.18
MNB-5	75.12 ± 2.08
MNB-6	83.25 ± 2.36
MNB-7	82.11 ± 2.54
MNB-8	79.76 ± 1.07

**Table 6 pharmaceutics-16-01451-t006:** Drug release kinetics of Dox.H-loaded nanohybrid SA-CS beads.

Kinetic Models		MNB-1	MNB-2	MNB-3	MNB-4	MNB-5	MNB-6	MNB-7	MNB-8
Zero-order kinetics	Ko	2.850	2.664	2.403	3.192	3.448	4.006	3.844	3.629
	R	0.8506	0.8740	0.8870	0.8808	0.8811	0.8747	0.8834	0.8824
First-order kinetics	K1	0.044	0.040	0.034	0.054	0.062	0.090	0.080	0.070
	R	0.9082	0.9216	0.9260	0.9406	0.9494	0.9646	0.9633	0.9553
Higuchi model	KH	12.154	11.290	10.100	13.585	14.700	17.236	16.459	15.527
	R	0.9325	0.9486	0.9565	0.9527	0.9521	0.9490	0.9540	0.9535
Korsmeyer–Peppas model	n	0.365	0.393	0.435	0.363	0.351	0.301	0.325	0.330
	R	0.9510	0.9616	0.9637	0.9681	0.9678	0.9713	0.9722	0.9716
Release kinetics		Fickian diffusion	Fickian diffusion	Fickian diffusion	Fickian diffusion	Fickian diffusion	Fickian diffusion	Fickian diffusion	Fickian diffusion

Note: rate constants (Ko, K1, KH), correlation coefficients (R), and the release exponent (n).

**Table 7 pharmaceutics-16-01451-t007:** Pharmacokinetics parameters of doxorubicin solution (oral) and doxorubicin-loaded Fa-GO hybrid mucoadhesive hydrogel beads (10 mg/kg dose).

Pharmacokinetic Parameter (Unit)	Dox.H	Fa-GO	t-Stat	*p*-Value
t1/2 h	518.4446	324.5803	313.8613	0.006
Tmax h	8	28	28	9.8995
Cmax μg/mL	63.31	163.22	168.82	41.9354
AUC 0-t μg/mLh	2084.87	7428.1	59.0881	4.8 × 10^−5^
AUC 0-t/0-inf_obs	0.264016	0.071258	0.280329	4.12188
MRT 0-inf_obs h	744.7376	440.2277	−0.1567	0.88959
Vz/F_obs (mg)/(μg/mL)	0.255641	0.166437	−0.5032	0.64519
Cl/F_obs (mg)/(μg/mL)/h	0.000342	0.000355	−0.9791	0.40406

Note. All values are shown as mean ± SD (*n* = 3). Abbreviations: t1/2 (half-life), Tmax (time to max concentration), Cmax (max concentration), AUC 0-t (area under the curve from 0 to t), MRT 0-inf_obs (mean residence time), Vz/F_obs (apparent volume of distribution), Cl/F_obs (apparent clearance). A *p*-value < 0.05 was considered significant.

**Table 8 pharmaceutics-16-01451-t008:** Parameters evaluated for fabricated nanohybrid SA-CS beads during different storage periods at 40 °C/75% RH.

Parameters	0 (Days)	15 (Days)	30 (Days)	90 (Days)	180 (Days)	Rate Constant (Day 1)	t (90%) Days
Particle size (µm)	341.0 ± 0.45	341.0 ± 0.72	340.7 ± 0.83	340.5 ± 0.32	319.1 ± 0.27	-	-
Drug constant (%)	100	100	99.87	99.12	98.77	8 × 10^−5^	1312.5

Note: All values are expressed as mean ± SD, where *n* = 3.

## Data Availability

Data are contained within the article.

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
