# Peer review of "Preparation and Evaluation of pH-Sensitive Chitosan/Alginate Nanohybrid Mucoadhesive Hydrogel Beads: An Effective Approach to a Gastro-Retentive Drug Delivery System"

_pharmaceutics, 2024, doi:10.3390/pharmaceutics16111451_

Round 1

Reviewer 1 Report

Comments and Suggestions for Authors

Sadia Rehman and co-worker presented an interesting study on Preparation and evaluation of pH-sensitive chitosan/alginate nanohybrid mucoadhesive hydrogel beads: An effective approch of gastro-retentive drug delivery system, which can be sent for approval of publication, however before doing so, a significant improvement required as suggested below,

1. Authors suggested to include results apart from size remaining others are superficially claimed. Try reduce the methodology and background informations.

2. Line no. 39 "Helicobacter pylori" suggested to italicize 

3. Line no. 68 "Creating" suggested to use technical term.

4. Line no. 74-76: "Because of the extended residence period and tighter contact between the formulation and the absorptive membrane, achieving mucoadhesion of a formulation in the digestive system may enhance the bioavailability of the drug entrapped within it", suggested to avoid starting a sentence using "Because", this usually come when authors don't use their own thinking and relay more on AI-based tools.

5. Line no. 81 "cheap cost" can't it be "economic"

6. Suggested to abbreviate nanoparticles "NPs", which have been used on several place.

7. Line no. 106. Very difficult to understand "Ugent Tech Sdn Bhd of Malaysia provided the graphene oxide (GO) for purchase.

8. Specific reagents were used in the fabrication of delivering system, thus authors suggested to indicate the chemicals purity and possibly mol.wt.

9. Suggested to supplement the in built arrangement for Ex vivo mucoadhesive analysis.

10. Line 270-271; Does withdrawal of blood required puncture of vein on different interval ? or a butterfly needle was used to puncture once and at stated interval blood was withdrawn. "Following dosing, 1 ml of blood was drawn from each rabbit's jugular vein at various intervals and placed in tubes with EDTA as an anticoagulant." Suggested to please elaborate the process clearly to help other researcher for reproducibility. 

 11. Suggested to indicate what was the yield of the fabrication, rather than directly switching to optical property.

12. Suggested to add the magnification value in the figure 

13. Authors are suggested for extension of hkl values from PXRD data obtained.

14. English editing required throughout the manuscript. 

Comments on the Quality of English Language

English editing required throughout the manuscript. 

Author Response

Response to Reviewer 1 

1: Authors suggested to include results apart from size remaining others are superficially claimed. Try reduce the methodology and background informations.

  • Response 1: Respected reviewer, thank you for your feedback. We appreciate your thorough review. Additionally, we express our gratitude for your insightful comments which have significantly improved the readability and overall quality of the manuscript. In response, we revised the manuscript to include detailed results, ensuring a comprehensive data presentation. Additionally, we condensed the methodology and background sections to streamline the content and focus more on key findings.
  1. Line no. 39 "Helicobacter pylori" suggested to italicize
  • Response 2: As commented by the respected reviewer, the correction has been made and updated in the revised manuscript. The change can be found in the Introduction section, line 39.
  1. Line no. 68 "Creating" suggested to use technical term.
  • Response 3: As per the respected reviewer's suggestion, the correction has been made, and the manuscript has been updated. In the Introduction (line 67), the word "creating" has been replaced with the technical term "development of".
  1. Line no. 74-76: "Because of the extended residence period and tighter contact between the formulation and the absorptive membrane, achieving mucoadhesion of a formulation in the digestive system may enhance the bioavailability of the drug entrapped within it", suggested to avoid starting a sentence using "Because", this usually come when authors don't use their own thinking and relay more on AI-based tools.
  • Response 4: As suggested by the reviewer, the sentence has been revised to avoid starting with "Because" and now reads more naturally. The updated lines are now 73-76 in the revised manuscript.
  1. Line no. 81 "cheap cost" can't it be "economic"
  • Response 5: As suggested by the worthy reviewer, we have revised the wording to improve clarity and formality. The phrase "cheap cost" has been updated to "economic" to better convey the intended meaning (line 80).
  1. Suggested to abbreviate nanoparticles "NPs", which have been used on several place.
  • Response 6: As the respected reviewer suggested, we have standardized the terminology by abbreviating "nanoparticles" as "NPs" throughout the manuscript, ensuring consistency in its usage.
  1. Line no. 106. Very difficult to understand "Ugent Tech Sdn Bhd of Malaysia provided the graphene oxide (GO) for purchase. "
  • Response 7: Thank you for your valuable feedback. We have revised the sentence for improved clarity. It now reads: "Graphene oxide (GO) was purchased from Ugent Tech Sdn Bhd, Malaysia" (updated Line 105).
  1. Specific reagents were used in the fabrication of delivering system, thus authors suggested to indicate the chemicals purity and possibly mol.wt.
  • Response 8: As recommended by the esteemed reviewer, the correction has been made, and the manuscript has been updated (Section 2.1. Materials, lines 105-116).
  1. Suggested to supplement the in-built arrangement for Ex vivo mucoadhesive analysis.
  • Response 9: As suggested by the esteemed reviewer, the correction has been made and updated in the revised manuscript. The mucoadhesive property of the hydrogel beads is now described in detail, with the corrected lines 278-2
  1. Line 270-271; Does withdrawal of blood required puncture of vein on different interval? or a butterfly needle was used to puncture once and at stated interval blood was withdrawn. "Following dosing, 1 ml of blood was drawn from each rabbit's jugular vein at various intervals and placed in tubes with EDTA as an anticoagulant." Suggested to please elaborate the process clearly to help other researcher for reproducibility.
  • Response 10: As per the esteemed reviewer's recommendation, we have clarified the blood withdrawal process to enhance reproducibility. The updated manuscript now states that a butterfly needle was inserted once, allowing for subsequent blood samples to be collected at the specified intervals without further vein punctures (lines 305-308).
  1. Suggested to indicate what was the yield of the fabrication, rather than directly switching to optical property.
  • Response 11: Thank you for your insightful feedback. We have incorporated the yield of the fabrication process to provide better clarity on the efficiency of the production method before addressing the optical properties. This revision has been applied in Section 3.1 (Percentage Yield) and Sections 331-340 (Results and Discussion).
  1. Suggested to add the magnification value in the figure 
  • Response 12: Thank you for your valuable feedback. As recommended, the magnification value has been added to the figure to enhance visualization and clarity. The changes can be seen in Figure 2: SEM image at 1000X and 10000X magnification of (a) Folic acid conjugated graphene oxide (Fa-GO), (b) MNB-1, (c) MNB-6, (d) drug loaded MNB-6.
  1. Authors are suggested for extension of hkl values from PXRD data obtained.
  • Response 13: We appreciate the reviewer's feedback. However, as discussed in the PXRD analysis included in the manuscript, the fabricated hydrogel beads exhibit an amorphous structure, as evidenced by the broad, diffused halo observed in the diffraction pattern. In contrast, the drug shows sharp peaks indicative of its crystalline nature, which is altered upon encapsulation within the polymeric matrix. Amorphous materials do not possess hkl values, as these correspond to the Miller indices that define specific crystallographic planes within a well-ordered crystal lattice. Bragg explained the diffraction of x-rays by crystals using a model in which the atoms of a crystal are regularly arranged in space and can be regarded as lying in parallel sheets separated by a definite and defined distance d. Since amorphous materials lack long-range periodic order, they do not exhibit well-defined crystal planes. Therefore, extending or assigning hkl indices to the PXRD data presented is not applicable. Hence, hkl value does not apply to crystal systems of lower symmetry [1, 2].
  1. English editing required throughout the manuscript. 
  • Response 14: The manuscript has been thoroughly revised in accordance with the esteemed reviewer's suggestions. It has been meticulously reviewed and corrected for grammatical errors to ensure clarity and precision.

References

  1. Sun, S., et al., Identification of the Miller indices of a crystallographic plane: a tutorial and a comprehensive review on fundamental theory, universal methods based on different case studies and matters needing attention. Nanoscale, 2020. 12(32): p. 16657-16677.
  2. Brittain, H.G., X-ray diffraction III: pharmaceutical applications. Spectroscopy, 2001. 16(7): p. 14-18.

Reviewer 2 Report

Comments and Suggestions for Authors

The manuscript addresses an important issue involving the development of a new possible treatment approach to gastro-retentive drug delivery system. Therefore, I recommend the publication of this manuscript. However, some concerns are highlighted:

1) In table 1, add the captions.

2) In item 3.1.2. Field emission SEM analysis:

Briefly detail how the sample was prepared for SEM analysis.

3) I am concerned about the encapsulation efficiency methodology. The author reports that the excess unloaded drug was removed (washed). However, immediately afterwards, the authors report that the unloaded drug was quantified. How did the authors quantify the unloaded drug if they removed it by washing? Please justify.

4) In the item In vitro drug release analysis: add collection times. Add controls (was free drug control performed?)

What was the composition of this buffer? I imagine that for a gastric retentive nanomaterial it must have followed the composition of the target site.

5) In the results section: add acronym captions to all tables and figures.

Add statistical difference in graphs and tables (indicate p-value in captions). Only results that can be statistically analyzed.

6) In section 4.10. Drug Release Study:

Please leave both graphs with the same y-axis, so that it will be possible to observe more clearly the difference in release between the two receptor media.

7) Add the limitations of this study.

Comments on the Quality of English Language

A new English correction needs to be done for better understanding.

Author Response

Response to Reviewer 2

  1. In table 1, add the captions.
  • Response 1: We appreciate the reviewer’s suggestion. The caption for Table 1 has been added in the updated manuscript.
  1. In item 3.1.2. Field emission SEM analysis: Briefly detail how the sample was prepared for SEM analysis.
  • Response 2: Using a scanning electron microscope, the surface morphology of drug-loaded Fa-GO, and (modal Nova Nano SEM). All the samples were freeze-dried to remove moisture while preserving their structure. The samples were mounted on an SEM stub and then coated with a thin layer of gold to enhance conductivity and minimize charging during the electron beam exposure. The coated samples were then examined using a modal Nova Nano SEM at magnifications of 1000X and 10000X, with an accelerating voltage set at 5.00 kV to obtain high-resolution images of the surface morphology [1].
  1. I am concerned about the encapsulation efficiency methodology. The author reports that the excess unloaded drug was removed (washed). However, immediately afterwards, the authors report that the unloaded drug was quantified. How did the authors quantify the unloaded drug if they removed it by washing? Please justify.

  • Response 3: We are grateful to the respected reviewer for the valuable comment. The detailed explanation of the encapsulation efficiency methodology is provided below. Additionally, this method has been well-established in previous literature [2-5].

After mixing doxorubicin (Dox.H) with the Fa-GO solution, the mixture was allowed to interact for 24 hours to enable the binding of Dox.H with Fa-GO. To remove any excess, unbound drug, the sample was washed. However, these wash solutions were not discarded; they were carefully collected for further analysis. The unbound doxorubicin in the wash solutions was quantified using a UV spectrophotometer at 480 nm. This step provided the necessary data to determine the amount of doxorubicin that had not been encapsulated by Fa-GO. By comparing the measured unbound drug to the initial amount of doxorubicin used, the encapsulation efficiency was calculated using the following formula:

  1. In the item In vitro drug release analysis: add collection times. Add controls (was free drug control performed?) What was the composition of this buffer? I imagine that for a gastric retentive nanomaterial it must have followed the composition of the target site.
  • Response 4: We appreciate the reviewer's valuable feedback. The in vitro drug release analysis section has been revised to include specific collection times and details of control studies. The composition of the buffers used (pH 1.2 and pH 5.6) was also clarified to reflect gastric conditions, ensuring accurate simulation of the target site. These updates can be found in lines 256-265 of the final manuscript.
  1. In the results section: add acronym captions to all tables and figures. Add statistical difference in graphs and tables (indicate p-value in captions). Only results that can be statistically analyzed.
  • Response 5: As recommended by the respected reviewer, comprehensive acronym captions have been added to all tables and figures to enhance clarity and ensure that readers can easily understand the terminology used throughout the results section.
  1. In section 4.10. Drug Release Study: Please leave both graphs with the same y-axis, so that it will be possible to observe more clearly the difference in release between the two-receptor media.
  • Response 6: As suggested by the worthy reviewer, both graphs are adjusted to share the same y-axis, allowing for a clearer and more straightforward comparison of drug release between the two receptor media. Changes can be seen in Figure 9: Percentage drug release study of doxorubicin-loaded nanohybrid SA-CS beads. Comparison of FTIR spectra of hydrogel beads before and after drug release.
  1. Add the limitations of this study.
  • Response 7: We acknowledge the reviewer's insight and have added a discussion on the study's limitations in the updated manuscript (lines 757-763). The limitations include reliance on in vitro assays that may not fully represent in vivo performance, limited long-term stability testing despite adherence to ICH guidelines, and the absence of thermal stability and cell line studies. We plan to address these aspects in future research for a comprehensive evaluation of the nanohybrid SA-CS beads.
  1. Comments on the Quality of English Language

A new English correction needs to be done for better understanding.

  • Response 8: The manuscript has been comprehensively revised as per the respected reviewer’s suggestions. All grammatical errors have been carefully reviewed and corrected using Grammarly Premium software to enhance the clarity and quality of the text.

Response to Reviewer 3

The present study focuses on i) the production of drug delivery vehicles based on folic acid functionalized graphene oxide nanoparticles and a hydrogel matrix, and ii) the study of the loading and release behavior of doxorubicin HCl on the novel delivery vehicles.

The planned work was carried out with sufficient consistency and the presentation of the results is understandable, clear and with sufficient references to the literature. A big plus is the use of only two self-citations.

I have no remarks on the present work and the study is recommended for publication

  • Response 1: Respected reviewer, thank you very much for your positive feedback and for recommending our study for publication. I appreciate your thoughtful review and am glad to hear that the presentation and referencing met your expectations.

References

  1. Khan, S., et al., Enhanced in vitro release and permeability of glibenclamide by proliposomes: Development, characterization and histopathological evaluation. Journal of Drug Delivery Science and Technology, 2021. 63: p. 102450.
  2. Sun, X., et al., Nano-graphene oxide for cellular imaging and drug delivery. Nano research, 2008. 1(3): p. 203-212.
  3. Chauhan, G., et al., “Gold nanoparticles composite-folic acid conjugated graphene oxide nanohybrids” for targeted chemo-thermal cancer ablation: in vitro screening and in vivo studies. European Journal of Pharmaceutical Sciences, 2017. 96: p. 351-361.
  4. Zainal-Abidin, M.H., M. Hayyan, G.C. Ngoh, and W.F. Wong, Doxorubicin loading on functional graphene as a promising nanocarrier using ternary deep eutectic solvent systems. ACS omega, 2020. 5(3): p. 1656-1668.
  5. Tiwari, S., P. Mistry, and V. Patel, SLNs based on co-processed lipids for topical delivery of terbinafine hydrochloride. J Pharm Drug Dev, 2014. 1(6): p. 604.

Reviewer 3 Report

Comments and Suggestions for Authors

The present study focuses on i) the production of drug delivery vehicles based on folic acid functionalized graphene oxide nanoparticles and a hydrogel matrix, and ii) the study of the loading and release behavior of doxorubicin HCl on the novel delivery vehicles.

The planned work was carried out with sufficient consistency and the presentation of the results is understandable, clear and with sufficient references to the literature. A big plus is the use of only two self-citations.

I have no remarks on the present work and the study is recommended for publication

Author Response

Response to Reviewer 3

The present study focuses on i) the production of drug delivery vehicles based on folic acid functionalized graphene oxide nanoparticles and a hydrogel matrix, and ii) the study of the loading and release behavior of doxorubicin HCl on the novel delivery vehicles.

The planned work was carried out with sufficient consistency and the presentation of the results is understandable, clear and with sufficient references to the literature. A big plus is the use of only two self-citations.

I have no remarks on the present work and the study is recommended for publication

  • Response 1: Respected reviewer, thank you very much for your positive feedback and for recommending our study for publication. I appreciate your thoughtful review and am glad to hear that the presentation and referencing met your expectations.

Round 2

Reviewer 1 Report

Comments and Suggestions for Authors

The authors have reflected all the said suggestions and comments, which made the manuscript enhanced with improved readability; Thus, I suggest for further consideration with acceptance.

Reviewer 2 Report

Comments and Suggestions for Authors

Thank you for considering my suggestions.